# Government influence on e-government adoption by citizens in Colombia: Empirical evidence in a Latin American context

**Juan Pablo Ramirez-Madrid** [1,2]*, **Manuela Escobar-Sierra**[2], **Isaias Lans-Vargas**[1], **Juan Manuel Montes Hincapie**[2]

1 Centre of Studies in Digital Government and Mobility, Quipux, Medellín, Colombia, 2 School of Economic and Administrative Sciences, Universidad de Medellín, Medellín, Colombia

* juan.ramirez@quipux.com

## Abstract

This study aims to identify government influence in the adoption of e-government by citizens (AEC) through a case study analyzing actions in Antioquia, Colombia, to increase AEC in annual vehicle tax filing and payment services. We classified these actions employing institutional theory, institutional interventions, and legitimation strategies. An analysis correlating AEC actions (including the COVID-19 containment measures) with over 16 million transactions in these two services during 2015–2020 found a strong government influence on AEC. We established coercive pressure and conformance to the environment as important predictors of AEC, but the COVID-19 containment measures only influenced electronic tax payments. Service type was also an essential predictor for these services; however, mobilization was not. Increasing AEC should be considered a necessary objective for public administrations, especially in developing countries that face shortages of resources and facilities.

**Data Availability Statement:** All relevant data are within the paper and its Supporting Information

## Introduction

The use of information and communication technologies (ICT) has transformed interactions with all government stakeholders [1], particularly with citizens [2]. The use of the Internet and other digital means to access government information and services has been defined as electronic government or e-government [3–5]. It emerged in the early 1990s and uses web-based IT as an important part of outreach to citizens [6, 7]. E-government has been a key catalyst in the transformation agenda of new public management for reinventing public administration and making it more efficient and effective [8, 9]. The value of e-government is recognized worldwide, and research has identified several benefits related to its implementation and use, such as transparency, efficiency, cost reduction, service delivery improvement, accountability, and reduction of corruption [8, 10–13]. The UN placed it at the center of the 17 Sustainable Development Goals for 2030 [14], and in recent years, e-government has demonstrated a positive global trend towards a higher level of development [15]. Nevertheless, the adoption of e-government by citizens (AEC) is crucial to achieving such benefits [11]. Indeed, e-government

files. Data also within https://doi.org/10.3886/E152221V1.

**Funding:** The author(s) received no specific funding for this work.

**Competing interests:** The authors have declared that no competing interests exist.

creates public value when citizens adopt it and also promotes private value acquisition for users, encouraging continuous use [16, 17].

According to Kumar et al. [18], AEC starts with citizens' decision to use online services (for example, choosing this channel over visiting the government office). The next stage of AEC involves increasing the frequency of use. However, various scientific studies indicate that AEC levels remain low [19, 20]. Savoldelli, Codagnone, and Misuraca [21] have referred to this as the e-government paradox, implying that public money and effort are wasted if investments in technology for e-government implementation do not translate into appropriate adoption responses from citizens.

Consequently, increasing AEC should be considered a necessary objective for public administrations. Therefore, this subject has been broadly studied, and various perspectives have been proposed [22, 23]. One perspective focused on the inhibitors of AEC [24], viewing non-adoption as an option for some citizen groups [25]. Another studied the digital divide's effects on the AEC process, acknowledging that not all citizens would benefit from such services as some may not have access to the Internet or do not know how to use such services [15, 26–28]. A third group examined channel choice and public service delivery, comparing e-government to traditional service delivery channels, including telephoning or visiting government offices [29].

Furthermore, Lamberti, Benedetti, and Chen [30] determined that citizens' benefits were essential determinants in choosing AEC over offline channels. Moreover, some studies on AEC for various services found that adoption levels have varied according to service type [10, 31, 32]. Finally, strong support has been found in 80% of AEC studies [24, 33, 34] for technology adoption theories and models, such as the Technology Acceptance Model (TAM) [35], the Unified Theory of Acceptance and Use of Technology (UTAUT) [36], the Diffusion of Innovation Theory (DOI) [37], and the Information System Success Model (ISS) [38].

Technology adoption theories and models have contributed to e-government, presenting the bases for implementing electronic services and pursuing adoption [24, 33, 34]. Thus, technology characteristics are strong determinants of AEC. Different technology aspects (usefulness, usability, look and feel, reliability, data quality, content quality, information availability, performance, integrity, efficiency, accessibility, compatibility, security, and privacy) are recognized as essential for persuading more citizens to use e-government actively [39–42]. However, AEC goes beyond technology adoption and relates to institutional and political issues; the relationship between the government and the citizens is also influential [21]. Government entities, the legislature, and the legal system are considered political institutions [43, 44], providing ways to reduce uncertainty and increase cooperation in the political arena [45] through guidelines for human action or appropriate behavior in society [44].

Consequently, despite its responsibility on the offer side, the government also has the potential to influence the demand side and increase AEC. For example, the E-Government Development Index, published by the United Nations [15], highlights the key role of governments, not only in increasing the services offered (the Online Service Index) but in improving the enabling conditions for AEC (the Telecommunications Infrastructure Index and the Human Capital Index). Also, Savoldelli et al. [21] noted shifts in the literature over time from technical and operational to institutional and political issues and found institutional and political barriers to be among the main factors affecting AEC. The supply of high-quality, attractive e-government services is the necessary but insufficient condition of use. Assuming citizens will use e-government services automatically once they are available is a serious mistake [46]; hence, government institutions must create an environment conducive to increasing AEC.

This study used an institutional framework as the theoretical basis. Considering the various pressures, developmental goals, and the existing social norms and beliefs favoring legitimacy

over efficiency that are relevant to the operationalization of e-government, an institutional perspective is appropriate for studying e-government initiatives and their adoption [47, 48]. Previous literature has similarly proposed institutional theory as a powerful lens to understand e-government [49, 50] and provided a framework to investigate the phenomenon in a wider context with multi-level analyses [48].

Institutional theory literature has identified three kinds of institutional pressures that can influence organizations and individuals: coercive, mimetic, and normative [51, 52]. King et al. [53] listed the institutions influencing IT innovation, highlighting the power of government entities in the adoption process (e.g., national government agencies, provinces, prefectures, states, and municipalities). Scott [52] recognized that states rely on institutional pressures to exert influence, and state actors are more likely to employ coercion. The legal framework, as an example of coercive pressure, is determinant in the context of AEC [54]. Several studies have confirmed that the government's entities use this coercive pressure to encourage citizens and other groups to embrace e-government [13, 49, 55–58], even in mandatory use contexts, as Denmark and the UK have defined [59]. Consequently, we have defined that coercive pressure predicts AEC as our first hypothesis.

Adoption and legitimacy are connected: as innovation spreads, adoption provides legitimacy [60], and from an institutional perspective, adoption becomes a way of demonstrating legitimacy [61]. Suchman [62] presented conformance to the environment as the easiest legitimacy acquisition strategy where the organization modifies internally to respond or satisfy the constituent tastes. Some studies have explored legitimacy's importance in e-government [47, 63–65]. For example, Trakhtenberg [66] proposed viewing AEC as the process of securing institutional legitimation for each of the government agencies whose services are offered by the e-government. Overall, we have defined that government conformance to the environment (citizens' needs) predicts AEC as our second hypothesis.

Furthermore, King et al. [53] described the institutional interventions (knowledge building, knowledge deployment, subsidizing, mobilization, standard-setting, and innovation directives) employed by various institutions to influence IT innovations. They built a model of potential institutional actions with two dimensions: the influence and regulation that institutions might exert and the "supply-push" and "demand-pull" forces that provide a context for those actions. Some e-government studies have referenced this model [67–69]. Governments can effectively employ institutional interventions to obtain desired behavioral changes and increase AEC [70, 71]. For example, mobilization has been recognized as an important element in AEC. Designing robust marketing campaigns to promote the benefits of e-government engagement can influence channel choice and behavior, guiding citizens to choose cost-effective channels to access e-government [13, 46]. Governments should run advertising campaigns informing people about the services to help develop positive attitudes towards e-government among citizens and the intention to use and the actual use of e-government services [3, 72, 73]. Indeed, good marketing campaigns are considered directly influential in promoting the intention to use e-government [59]. Consequently, we have defined that mobilization predicts AEC as our third hypothesis. Also, we defined that knowledge deployment predicts AEC as our fourth hypothesis.

Finally, some studies focused on cross-sectional analysis and suggested longitudinal analysis in future research [2, 74]. For example, Hofmann, Räckers, and Becker [22] recommended longitudinal studies to eliminate short-term effects and understand the e-government domain development. Therefore, this research used a case study in Colombia, a developing Latin American country. We analyze the government actions to increase AEC in the annual vehicles' tax filing and payment (offered by the same government entity using the same physical and electronic channels), correlating with the adoption behavior from 2015 to 2020. Additionally,

we analyzed the impact of the measures implemented in 2020 to contain the COVID-19 pandemic on AEC. Our research question is: how does government influence AEC? We built our hypothesis using institutional theory [51], institutional interventions [53], and legitimation strategies [62]. Studying AEC in developing countries is essential for providing theoretical and practical contributions to the literature [23, 75].

The following section presents the methodology, followed by a section on the results, discussion, and conclusions.

## Methodology

### The case study

Colombia is a Latin American developing country located in South America's northern region and has over 50 million inhabitants, a capital district, and 32 government departments. According to ICT Ministry of Colombia [76], 64% of the population has Internet access at home, and 74% has access to a smartphone. However, 97% of the population mainly uses the Internet to communicate and interact—90% use it daily, while 27% utilize it for transactions. Additionally, social media platforms are preferred among the population—88% use Facebook, 87% use WhatsApp, 48% use YouTube, 34% use Instagram, and 20% use Twitter.

Latin American governments are emulating the actions taken years ago by developed countries and need to move towards more locally-tailored technologies to enhance e-government [77]. For example, Acosta, Acosta-Vargas, and Lujan-Mora [78] found an absence of compliance with the Web Content Accessibility Guidelines (WCAG) 2.0 among websites of Latin American countries that offer e-government services, exacerbating digital divide consequences. Consequently, the Colombian government has applied different strategies to improve e-government. For example, on May 25, 2019, the government issued Law 1955, as part of the National Development Plan 2018–2020, which defined digital government in terms of institutional management and performance policy; alongside this, the Digital Government Direction was created to coordinate all government entities' efforts [79]. Complementarily, in July 2021, the Colombian government enacted Law 2108, declaring the Internet an essential and universal public service [80].

Colombia was the Latin American leader in the UN e-government survey [81] in 2010 due to the unique strategies of its government. The Organization for Economic Co-operation and Development (OECD) placed Colombia third in its 2019 Digital Government Index [82]. Similarly, in the national Digital Government Performance Report for 2018, Colombia's sixth-largest department, Antioquia (capital: Medellín), had the second-best performance [83]. The UN E-Government Survey for 2020 also accords a high e-government Development Index (EGDI) of 0.7164 to Colombia. Other countries in the region are also progressing rapidly; Argentina, Brazil, Chile, and Costa Rica significantly improved their EGDI values between 2018 and 2020 and transitioned to the "very high EGDI" group [15]. Among the three components of Colombia's EGDI (human capital, online services, and telecommunication and infrastructure), the first two elements ranked very high, but telecommunications and infrastructure still need attention.

Our case study centers on the Department of Antioquia's annual vehicle tax filing and payment services, which are available via a website (www.vehiculosantioquia.com), a mobile application, and physical offices. The department has total control for these services and can issue regulations ruling the process, the service design, technology selection, and the channels through which the services are offered. For 2020, the Department of Antioquia had 1.5 million vehicles, and approximately 1.2 million are subject to this tax (motorcycles with engines smaller than 125 cm$^3$, public transport vehicles, and government vehicles are exempt). The

**Table 1. Annual adoption of e-filings and e-payments.**

| Year | Total filings | E-filings | Adoption | Total payments | E-payments | Adoption |
|---|---|---|---|---|---|---|
| 2015 | 1,365,811 | 707,935 | **51.83%** | 678,674 | 90,484 | **13.33%** |
| 2016 | 1,350,360 | 703,167 | **52.07%** | 730,165 | 92,290 | **12.64%** |
| 2017 | 1,658,132 | 1,070,757 | **64.58%** | 805,175 | 140,307 | **17.43%** |
| 2018 | 1,778,395 | 1,203,783 | **67.69%** | 767,274 | 159,456 | **20.78%** |
| 2019 | 2,081,664 | 1,490,005 | **71.58%** | 949,449 | 268,346 | **28.26%** |
| 2020 | 3,001,148 | 2,369,015 | **78.94%** | 1,076,093 | 443,728 | **41.24%** |
| TOTAL | **11,235,510** | 7,544,662 | **TOTAL** | **5,006,830** | 1,194,611 | |

annual payment rate is 79%, with the remaining 21% being defaulters. Almost 25% of the department's annual income comprises overdue amounts, including penalties and interest.

This case study refers to the significant increase in AEC between 2016 and 2020 for these two services (see Table 1). The adoption of tax filing e-service rose from 52.07% to 78.94%, representing an increase of 51.60%. Additionally, the adoption of tax payment e-service went from 12.64% to 41.24%, representing an increase of 226.27%. The objective is to study AEC between 2015 and 2020 to identify the actions performed by the government to promote AEC and determine which of these actions could be used as AEC predictors. This case study is relevant because AEC is generally low in Latin America, and the adoption of these services stands out among regional adoption rates. Furthermore, studies revealed that approximately 7% of citizens have attempted to use e-government services, with only 4% conducting 100% online [84, 85].

Tax payment requires a previous tax filing, and e-payments require e-filings. Multiple filings for one payment are typical. Also, having filings without payment is usual. For example, a tax filing to pay with discounts should be updated if not paid before the deadline. Another example is presented when the vehicle's information should be updated (e.g., the new owner or updated value). Thus, the payment is performed based on the last tax filing. It is impossible to pay through e-government channels once the tax is filed physically in an office, and this means a selection process affects e-payments defined by the filing presented by the electronic channels.

Additionally, it is vital to mention that the national government generates information every year to configure the basis for tax calculation. This configuration is performed in the first weeks of each year, during which the system is inaccessible, but some manual filing may still be generated. Additionally, the government currently closes and balances the previous year's records.

## Data and methods

This is an explanatory case study [86, 87] that used interviews and documentation analyses combined with quantitative analyses of the historical data from 2015 to 2020 (more than 11 million filing records and more than 5 million payments, as illustrated in Table 1) to identify how the government influences AEC (our dependent variable). We employed descriptive methods, including aggregated results for the use of e-government services in graphical and tabular output [88, 89]. Additionally, we conducted multiple regressions for the data correlation analysis. Similar approaches have been used in previous research related to AEC [90, 91].

We conducted interviews to gather the information, verify data quality, interpret the information, and discuss and confirm the findings [86]. This method provides an opportunity for interaction and dialogue between the interviewees and interviewer that can be used to clarify, explore, and raise new issues [92, 93]. We prepared a resume for every interview validated in the following interview. Additionally, we created a repository to consolidate all the

information we gathered. In summary, we conducted 27 interviews from 2018 to 2021 with the general manager (GM), the operational manager (OM), and two data analysts (DA1, DA2) in charge of strategic and operational activities. We explained the study's objective and focused on defining adoption in the initial five interviews. We concluded that our AEC measurement would be the number of services requested in the e-government channel divided by the total number of services requested (e-payments/total payments and e-filings/total filings). Other alternatives analyzed were based on the total amount paid by channel or the number of citizens accessing the services by channel (a citizen may request several services by different channels). However, we decided that number of services was the best option for this research to avoid dealing with personal data. Then, we worked on the data analysis (how to organize the data and connect actions with the history of adoption) in 16 sessions. We obtained authorization to access the data for this research for ethical considerations, and the data provided did not include sensitive or personal information.

Further, we collected data from physical records and documents, digital records, and relevant laws and regulations. We analyzed this information to identify all actions related to the services that might predict AEC. Each action performed was classified according to the institutional factor of our hypothesis (coercive pressure, conformance to the environment, mobilization, and knowledge deployment) to be used as independent variables (see next section and S1 Appendix). Next, we analyzed the available data from 2015 to 2020, and we found that a weekly consolidation was suitable for our study. We initially executed daily analysis, but we identified some problems because physical offices were closed on weekends and holidays. Thus, all services were accessed only by electronic channels, making adoption 100% with low transactions for those days. Consequently, we analyzed the weekly consolidated data, represented over 313 weeks, including service type (e-filings, e-payments), week (represented by the Sunday dates), and the number of services per channel. We studied the behavior of the weekly AEC rate and worked on defining how to relate the institutional factors identified in the previous step with the AEC rate behavior.

Finally, we used six sessions to discuss and confirm the results and prepare the report. All the data from different sources support the credibility of the findings as they allow triangulation and capture contextual complexity [94].

## Government actions performed to increase AEC

The first action we identified was the alliance signed by the government in 2017 with a govtech company (IE University, 2020) [95], which focused on service improvement. As a result, an imported system (SAP) was integrated with an indigenous system developed by the govtech company. This type of action is recognized as knowledge deployment [53]. Additionally, we found other actions influencing the services presented in S1 Appendix.

First, the government establishes two payment due dates every year. The first date allows payment due at a 10% discount, and the second date is for the amount due without penalties, but interest accrues after this date. We also found some laws and regulations establishing special discounts in specific periods. Initial analyses showed that citizens that pay with discounts have higher AEC levels than citizens that pay with penalties. Also, we found that weeks with due dates have presented an increase in AEC, possibly since physical offices cannot receive the demand in these periods. Consequently, we classified these actions as coercive pressure [51], defining two variables: *Payment* and *laws and regulations*. We used the variable *payment* to establish the value of paying; 0.8 for the 20% discount period, 0.9 for the 10% discount period, 1.0 for the no-discount period, and 1.0 plus monthly interest rate for the weeks after the due date. For the variable *laws and regulations*, we used 1 if there was a deadline date in that week and 0 if it did not.

**Table 2. Survey for citizens' feedback on e-government services from 1 January 2018 to 10 March 2021.** Question one and two.

Q1: Tell us the degree of satisfaction obtained when using our platform.

| Year | Excellent | % | Good | % | Acceptable | % | Deficient | % | Total answers |
|---|---|---|---|---|---|---|---|---|---|
| 2018 | 1.100 | 51.0 | 681 | 31.6 | 193 | 8.9 | 183 | 8.5 | 2.157 |
| 2019 | 35.122 | 63.5 | 14.519 | 26.2 | 3.404 | 6.1 | 2.307 | 4.2 | 55.352 |
| 2020 | 39.733 | 65.4 | 15530 | 25.6 | 3496 | 5.8 | 1994 | 3.3 | 60.753 |
| 2021 | 16.789 | 71.9 | 5279 | 22.6 | 861 | 3.7 | 416 | 1.8 | 23.345 |

Q2: Rate how easy it was to carry out your procedure when accessing our platform and using its tools.

| Year | Excellent | % | Good | % | Acceptable | % | Deficient | % | Total answers |
|---|---|---|---|---|---|---|---|---|---|
| 2018 | 1.061 | 49.2 | 849 | 39.4 | 167 | 7.7 | 80 | 3.7 | 2.157 |
| 2019 | 34.507 | 62.3 | 17479 | 31.6 | 2286 | 4.1 | 1080 | 2.0 | 55.352 |
| 2020 | 38.407 | 63.2 | 18392 | 30.3 | 873 | 1.4 | 3081 | 5.1 | 60.753 |
| 2021 | 17.041 | 73.0 | 5070 | 21.7 | 865 | 3.7 | 369 | 1.6 | 23.345 |

Second, we found several technical solution updates to the e-services since 2017. We analyzed the objectives of these updates, and they refer to software improvements, mainly for security, privacy, usability, information quality, and new functionalities. We classified this software update as conformance to the environment [62]. To define an independent variable, we used a cumulative variable that added one to the value for every software update in the corresponding week. The objective was to reflect the solution's maturity accumulated over time.

Third, we identified different promotion campaigns starting in 2018. These campaigns were executed using SMS and social media advertising. We classified these actions as mobilization [53]. To define an independent variable, we consolidated the campaigns for every week, adding records promoting e-government and subtracting records promoting physical offices. We then normalized the values for this variable.

Fourth, we found that during the COVID-19 pandemic in 2020, the government (at national, regional, and local levels) undertook virus containment measures limiting physical offices services for many weeks. We classified these actions as coercive pressure [51] and defined the variable COVID-19. For this variable, we used 1 for the weeks the lockdown lasted and 0 for the other weeks.

Furthermore, we identified other actions to manage AEC. For example, we found evidence that permanent staff training sessions help improve service. Also, we found that monitoring of satisfaction, ease, and effectiveness of use and effective implementation of e-government services has been in place since November 2018. Additionally, after every transaction, the system requests the completion of a voluntary three-question survey—Tables 2 and 3 present the survey questions and the consolidated results, respectively.

The above actions are important because service quality (e.g., technical or output quality, functional or process quality, and direct customer service from employees) has been recognized as a strong predictor of AEC, and the recurring use of e-government will increase confidence in

**Table 3. Survey for citizens' feedback on e-government services from 1 January 2018 to 10 March 2021.** Question three.

Q3: Did you manage to complete your procedure promptly?

| Year | Yes | % | No | % | Total answers |
|---|---|---|---|---|---|
| 2018 | 1.680 | 77.9 | 477 | 22.1 | 2.157 |
| 2019 | 48.240 | 87.2 | 7112 | 12.8 | 55.352 |
| 2020 | 55.073 | 90.7 | 5680 | 9.3 | 60.753 |
| 2021 | 22.282 | 95.4 | 1063 | 4.6 | 23.345 |

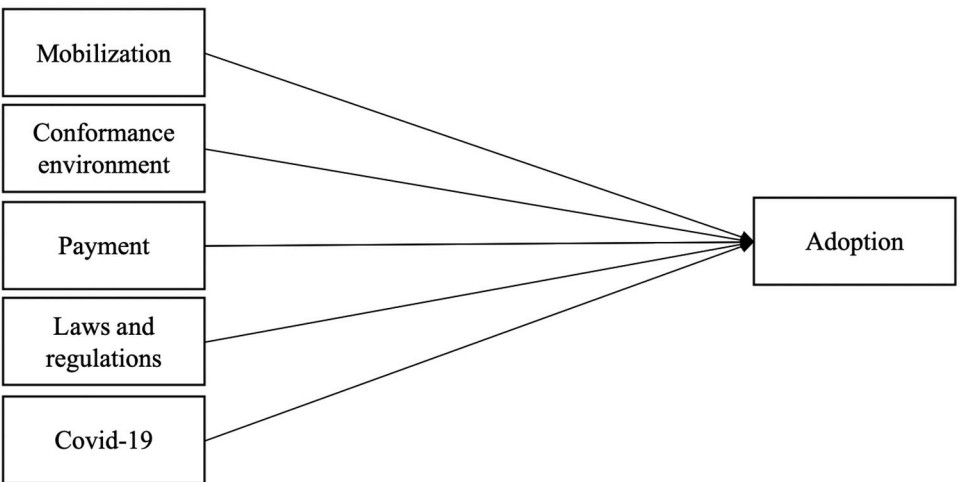

**Fig 1. Adoption model based on the findings and the information available for analysis.**

AEC if citizens perceive better customer service [96–98]. Complementarily, technical quality plays an essential role, influencing citizens' intentions to use e-government [2, 42, 99, 100]. Therefore, we classified these actions as conformance to the environment [62]. However, we did not identify sufficient action to implement an independent variable for the statistical analysis.

In summary, we observed actions for knowledge deployment, conformance to the environment (fulfilling demand-side expectations with software updates, quality of service, and quality of software), mobilization, coercive pressure (laws and regulations for due dates, discounts and penalties, and measures in 2020 related to COVID-19). After analyzing these actions and the available information, we defined the dependent variable and five independent variables to propose a model for the statistical analysis of AEC (Fig 1). For the independent variables, we added columns in the dataset according to the explanation given in this section.

## Results

### Exploratory analysis

**Descriptive analysis.**   We analyzed tax filings and tax payments separately. Fig 2 presents the annual behavior of AEC using the data from Table 1. For both services, we found differences between the AEC rate in the no-management period (2015–2017) and the AEC in the

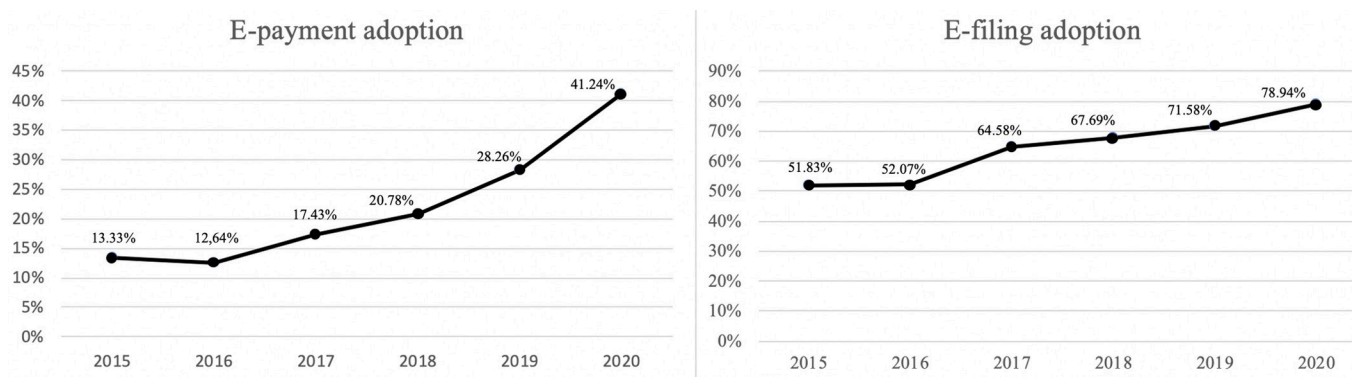

**Fig 2. AEC behavior for e-payments and e-filings from 2015 to 2020.**

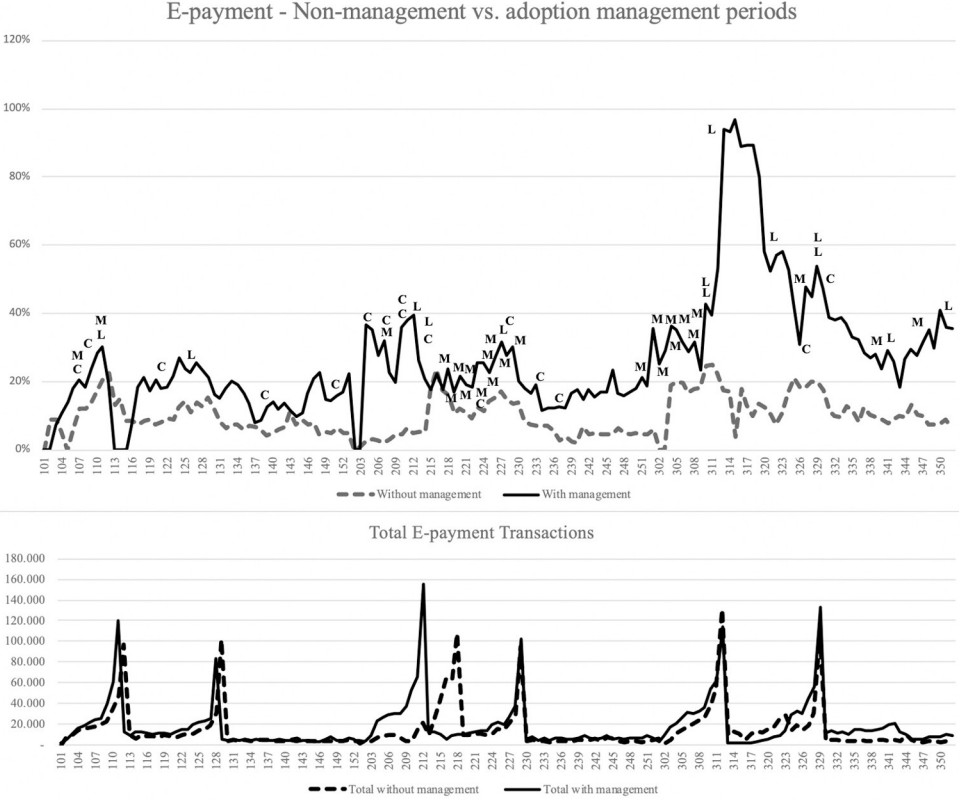

**Fig 3. E-payments: Comparing non-management (2015–2017) vs. adoption-management (2018–2020) periods—conformance to the environment (C), mobilization (M), and laws and regulations (L).**

managed period (2018–2020). Without AEC management, the adoption rate can decrease, as it did for e-payments during 2015–2016.

Figs 3 and 4 illustrate the weekly volume of services requested (bottom of the graph) and the AEC rate (top of the graph). The X-axis represents the 52-week year with the year (first digit, 1 to 3) and the week (01 to 52). Initial analysis identified six peaks in volume, two per year, with a notable increase in volume during the weeks preceding the due dates. These peaks in volume also represent AEC's increase, probably because the capacity of physical offices was overloaded, and citizens should use virtual channels. Furthermore, AEC for the first due date (payment with discount) was higher than the second due date from 2015 to 2019; for 2020, the second due date was during the pandemic contention period, and AEC was higher than the first. This might indicate that citizens that pay early, with discounts, tend to adopt more electronic channels.

Figs 3 and 4 also contain government actions that may have helped enhance AEC. Having observed adoption behavior for six years, we identified historic highs in adoption in 2020, especially during the pandemic-induced lockdown from late March (week 312) to July (week 330). Additionally, we found the service behavior shaped by the due dates that also seemed to predict AEC; the most significant adoption rate changes are associated with these dates. Fig 4 illustrates higher volumes of transactions observed in e-filings than in e-payments. For example, in week 30 of 2020 (third year of the second period, 330), more than 300,000 filings were performed, compared to less than 130,000 payments previously. Finally, we observed different changes in AEC related to mobilization and conformance to the environment activities since

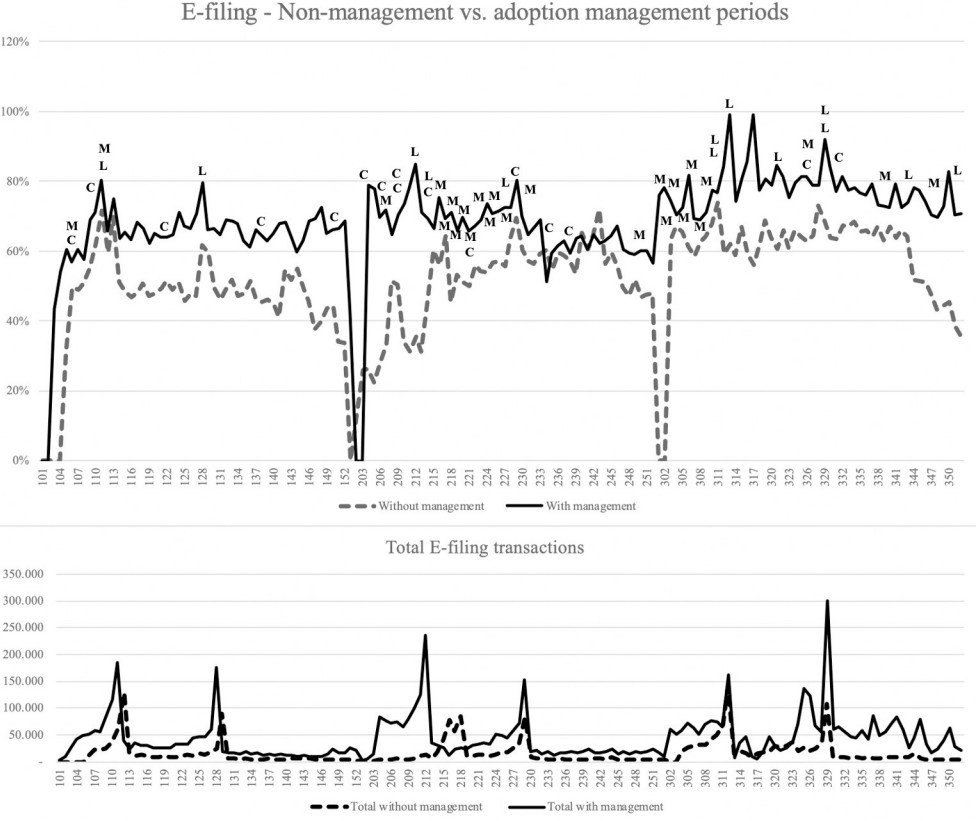

**Fig 4. E-filings: Comparing non-management (2015–2017) vs. adoption-management (2018–2020) periods—conformance to the environment (C), mobilization (M), and laws and regulations (L).**

2018. For example, we identified several mobilization campaigns, especially in 2019 and 2020, increasing the volume of services requested and AEC weeks before the due dates. However, it was not easy to find trends from this type of analysis.

## Statistical analysis

We conducted different regression analyses considering the adoption model proposed in Fig 1 and the available data. First, we started with ordinary least squares (OLS) regression, as we will present the results in Tables 4 and 5.

We identified in Table 4 that, from the five variables, only mobilization did not predict AEC for e-payment with a p = 0.636. However, OLS regression has favorable properties if its

**Table 4. OLS regression results (dependent variable: adoption of e-payment).**

| Predictor | Coefficient | Std. error | z | P>\|z\| | [0.025 | 0.975] |
|---|---|---|---|---|---|---|
| Constant | 21.8467 | 2.421 | 9.024 | 0.000 | 17.083 | 26.611 |
| Mobilization | -1.2302 | 2.595 | -0.474 | 0.636 | -6.337 | 3.877 |
| Conformance to the environment | 1.5594 | 0.127 | 12.307 | **0.000** | 1.310 | 1.809 |
| Payment | -10.4630 | 2.060 | -5.078 | **0.000** | -14.517 | -6.409 |
| Laws and regulations | 6.7654 | 2.523 | 2.682 | **0.008** | 1.802 | 11.729 |
| COVID-19 | 26.2523 | 3.173 | 8.274 | **0.000** | 20.009 | 32.495 |

N = 313; $R^2$ = 0.722; Adj. $R^2$ = 0.717; F = 159.2; p = 0.000.

**Table 5. Regression results (dependent variable: adoption of e-filing).**

| Predictor | Coefficient | Std. error | z | P>|z| | [0.025 | 0.975] |
|---|---|---|---|---|---|---|
| Constant | 49.3419 | 4.072 | 12.118 | 0.000 | 41.330 | 57.354 |
| Mobilization | 2.6855 | 4.365 | 0.615 | 0.539 | -5.903 | 11.274 |
| Conformance to the environment | 1.9364 | 0.213 | 9.087 | **0.000** | 1.517 | 2.356 |
| Payment | 1.8944 | 3.465 | 0.547 | 0.585 | -4.924 | 8.713 |
| Laws and regulations | 14.9517 | 4.242 | 3.524 | **0.000** | 6.604 | 23.300 |
| COVID-19 | -3.7885 | 5.336 | -0.710 | 0.478 | -14.288 | 6.711 |

N = 313; $R^2$ = 0.345; Adj. $R^2$ = 0.335; F = 32.4; p = 0.000.

assumptions are met but can give misleading results if those assumptions are not met. Thus, OLS is not robust to violations of its assumptions (normality and homoscedasticity). In this regression, we found that errors were not normally distributed across the data (Prob. Omnibus = 0.000) and heteroscedasticity in the variance of the errors across the dataset (Durbin-Watson: 0.639). This situation is typical of regressions applied to time series data, like our case.

Similarly, in the results presented in Table 5, we identified that mobilization, payment, and COVID-19 did not predict AEC for e-filing. However, we also found that errors were not normally distributed across the data (Prob. Omnibus = 0.000) and heteroscedasticity in the variance of the errors across the dataset (Durbin-Watson: 0.517). Consequently, we executed robust OLS for heteroscedasticity and autocorrelation consistency (HAC) regression. Robust regression methods are designed to be not overly affected by violations of the assumptions. Results will be presented in Tables 6 and 7.

The results presented in Table 6 identified that, from the five variables, only mobilization did not predict AEC for e-payment with a p = 0.726. Similarly, in the results presented in Table 7, we identified that mobilization, payment, and COVID-19 did not predict AEC for e-filing.

Complementarily, we expanded the analysis with a different regression model to verify previous results. Thus, we conducted a generalized linear regression (GLS): a Tweedie family distribution configured as a Poisson and Gamma distribution compound. In the GLS, errors can follow any distribution of the exponential family, and homoscedasticity is not essential for the distribution of the errors. Consequently, Table 8 presents the GLS regression results for AEC e-payment—conformance to the environment, payment, and laws and regulations measures predicted AEC for e-payment, but mobilization did not. This regression identified a variation with the COVID-19 variable; its p = 0.066 is above but close to the 5% limit. However, considering previous results, we claim that the COVID-19 variable should continue as a relevant predictor of AEC for e-payment.

**Table 6. Robust regression results (dependent variable: adoption of e-payment).**

| Predictor | Coefficient | Std. error | z | P>|z| | [0.025 | 0.975] |
|---|---|---|---|---|---|---|
| Constant | 21.8467 | 2.311 | 9.452 | 0.000 | 17.299 | 26.395 |
| Mobilization | -1.2302 | 3.508 | -0.351 | 0.726 | -8.133 | 5.673 |
| Conformance to the environment | 1.5594 | 0.135 | 11.589 | **0.000** | 1.295 | 1.824 |
| Payment | -10.4630 | 1.762 | -5.939 | **0.000** | -13.929 | -6.996 |
| Laws and regulations | 6.7654 | 2.135 | 3.169 | **0.002** | 2.565 | 10.966 |
| COVID-19 | 26.2523 | 6.877 | 3.817 | **0.000** | 12.720 | 39.785 |

N = 313; $R^2$ = 0.722; Adj. $R^2$ = 0.717; F = 64.07; p = 0.000; Covariance Type: HAC.

**Table 7. Robust regression results (dependent variable: adoption of e-filing).**

| Predictor | Coefficient | Std. error | z | P>|z| | [0.025 | 0.975] |
|---|---|---|---|---|---|---|
| Constant | 49.3419 | 6.218 | 7.935 | 0.000 | 37.106 | 61.578 |
| Mobilization | 2.6855 | 1.903 | 1.411 | 0.159 | -1.059 | 6.430 |
| Conformance to the environment | 1.9364 | 0.231 | 8.378 | **0.000** | 1.482 | 2.391 |
| Payment | 1.8944 | 4.514 | 0.420 | 0.675 | -6.988 | 10.777 |
| Laws and regulations | 14.9517 | 2.824 | 5.295 | **0.000** | 9.395 | 20.508 |
| COVID-19 | -3.7885 | 3.259 | -1.162 | 0.246 | -10.202 | 2.624 |

N = 313; $R^2$ = 0.345; Adj. $R^2$ = 0.335; F = 51.44; p = 0.000; Covariance Type: HAC.

Table 9 shows the GLS regression results for e-filing. From the five variables, only laws and regulations and conformance to the environment were shown to predict AEC, while mobilization, payment, and COVID-19 actions did not predict this variable.

In summary, considering the results presented in this section, we found that the laws and regulations and conformance to the environment are important predictors of AEC in the two services. We did not find predictive power for mobilization activities, but we found that payment and COVID-19 actions were essential AEC predictors for e-payments but not e-filings. Thus, factors predicting AEC also relate to the type of service. It is important to mention that all regression models had equivalent results identifying which factors predicted AEC, despite the coefficients and the standard errors being different. We highlight that the COVID-19 variable was present only in 2020 but became a relevant predictor for e-payment.

## Discussion

Considering the results presented in the last section, we consolidated the status of every hypothesis for the two analyzed services in Table 10. As a result, only H2 was confirmed for both services, H1 was partially confirmed, H3 was not confirmed, and we could not get enough information for statistical analysis for H4.

The predictive power of coercive pressure in AEC indicates that government entities possess an important tool for AEC management. Coercive pressure is represented in the laws and regulations establishing due dates (laws and regulations), discounts and penalties (payment), and access restrictions to physical offices during the COVID-19 lockdown in 2020, making e-government use mandatory [51]. Only laws and regulations predicted e-filings and e-payments; neither payment nor COVID-19 variables predicted e-filings. After analyzing the behavior and adoption levels of e-filings, we consider that this service has reached a high level

**Table 8. GLS regression results (dependent variable: adoption of e-payment).**

| Predictor | Coefficient | Std. error | z | P>|z| | [0.025 | 0.975] |
|---|---|---|---|---|---|---|
| Constant | 3.1506 | 0.129 | 24.437 | 0.000 | 2.898 | 3.403 |
| Mobilization | -0.0093 | 0.126 | -0.074 | 0.941 | -0.256 | 0.237 |
| Conformance to the environment | 0.0923 | 0.006 | 14.369 | **0.000** | 0.080 | 0.105 |
| Payment | -0.7490 | 0.110 | -6.786 | **0.000** | -0.965 | -0.533 |
| Laws and regulations | 0.4064 | 0.126 | 3.222 | **0.001** | 0.159 | 0.654 |
| COVID-19 | 0.2815 | 0.153 | 1.837 | 0.066 | -0.019 | 0.582 |

Notes: Model: GLM; Model Family: Tweedie (var_power = 1.8, meaning for a compound of Poisson and Gamma); Link Function: log; Method: IRLS; No. Iterations: 12; Covariance Type: nonrobust; No. Observations: 313; Df Residuals: 307; Df Model: 5; Scale: 0.33690; Log-Likelihood: nan; Deviance: 250.85; Pearson chi2: 103.

**Table 9. GLS regression results (dependent variable: adoption of e-filing).**

| Predictor | Coefficient | Std. error | z | P>|z| | [0.025 | 0.975] |
|---|---|---|---|---|---|---|
| Constant | 3.8750 | 0.076 | 51.192 | 0.000 | 3.727 | 4.023 |
| Mobilization | 0.0407 | 0.079 | 0.515 | 0.607 | -0.114 | 0.196 |
| Conformance to the environment | 0.0327 | 0.004 | 8.366 | **0.000** | 0.025 | 0.040 |
| Payment | 0.0582 | 0.064 | 0.905 | 0.365 | -0.068 | 0.184 |
| Laws and regulations | 0.2445 | 0.077 | 3.174 | **0.002** | 0.094 | 0.396 |
| COVID-19 | -0.0967 | 0.096 | -1.003 | 0.316 | -0.286 | 0.092 |

Notes: Model: GLM; Model Family: Tweedie (var_power = 1.8, meaning for a compound of Poisson and Gamma); Link Function: log; Method: IRLS; No. Iterations: 12; Covariance Type: nonrobust; No. Observations: 313; Df Residuals: 307; Df Model: 5; Scale: 0.15754; Log-Likelihood: nan; Deviance: 288.36; Pearson chi2: 48.4.

of adoption, close to 80% for 2020, which may be hard to increase, and the predictors seem to be different. Future research could study the maximum level of adoption of different e-government services and the predictors of services with high levels of AEC. For example, Becker et al. [28] studied the different levels of AEC compared to the total population, users of the Internet, and users of e-commerce.

These results align with those in the literature. Coercive pressures can be enablers or constraints in the AEC process [55, 58]. For example, some activities constrained AEC, including those related to due dates in the administrative collection available only in physical offices. The legal framework is important in shaping government services [54, 101]. Mandatory use was presented by Ghareeb et al. [59] as the strategy implemented for Denmark and the UK to force citizens to explore e-services and their benefits. This occurred during the COVID-19 lockdown; this variable was present only in 2020, when only the electronic channels were available, forcing new citizens to use the e-service, becoming an important predictor for e-payment. However, it was only effective in technically developed regions. Consequently, caution is required because the digital divide is an important issue in the Latin American region [15, 27]. Latin American governments could explore the gradual implementation of mandatory use, for example, starting with some services (e.g., accessing information), geographical areas (e.g., IT developed areas), or groups (e.g., public servants, students, teachers, and professors).

Indeed, Latin American governments have an essential role in combating the factors that widen or retain the digital divide, especially in rural areas and developing countries [102]. Increasing access to computers and the Internet (as the Colombian government promotes) is not a complete solution but a good start nonetheless [103]. The Internet can promote citizenship and citizen participation in Latin America [104]. Also, improving websites' accessibility to avoid discrimination, even imposing sanctions for non-compliance of standards as another example of coercive pressure, could be another strategy followed by developing countries [78].

We also found conformance to the environment to be an important predictor of AEC for both services. Improving technical solutions and monitoring satisfaction, ease of use, and effectiveness can help increase conformance with citizens' expectations of the technical

**Table 10. Hypothesis summary.**

| Hypothesis | e-payments | e-filings |
|---|---|---|
| H1. Coercive pressure predicts AEC | Confirmed | Partially confirmed |
| H2. Conformance to the environment (citizens' needs) predicts AEC | Confirmed | Confirmed |
| H3. Mobilization predicts AEC | Not confirmed | Not confirmed |
| H4. Knowledge deployment predicts AEC | NA | NA |

solution quality and service quality [62]. We identified that the maturity of electronic services, represented in conformance to the environment variable in the model, supports the annual increase of AEC we observed in the analysis. We found that the literature highlights the important influence on AEC of quality of service and technical quality [96–98], but this is the first study to classify these as part of a legitimation strategy. We concluded that no law, regulation, or mobilization could influence AEC increase without improving the quality of technical solutions and services. Additionally, governments should take action to fulfill the needs of AEC in developing countries in a broader form. For example, creating the legal framework that supports AEC and removes legal barriers, investing in creating reliable databases, enhancing interoperability among government agencies, investing in education and training, combating the digital divide, and improving technical aspects of the systems to make them easy to use for all audiences [15, 103, 105].

We found that mobilization was not significant for any service. We identified results similar to those of Henriksen and Damsgaard [70], where the demand-pull-based approach of the Danish government was successful, as was changing the strategy for imperatives and regulations. However, many studies have highlighted mobilization's importance in AEC [3, 72, 73]. We consider this result related to the nature of the services studied. This study looked at yearly, mandatory services, and most users would be recurrent users that know these services well. In this case, mobilization might impact other variables as early payments and increasing volume of transactions but not for AEC. Promotion campaigns are considered essential in the early stages of adoption, creating awareness among citizens and influencing the intention to use [59]; for example, in developing countries, promotion campaigns should leverage the social characteristics of the society [106, 107]. Additionally, the mobilization of services studied was mainly through electronic channels (SMS, social media advertising, web portal messages, and mobile applications), and non-adopters scarcely use these electronic channels. Therefore, conducting mobilization by combining traditional channels and electronic channels would be valuable for engaging non-adopters [11].

Although we did not have enough information for statistical analysis, we found evidence of the importance of knowledge deployment in AEC in three actions performed by the government: delegating services operation to experts, training public servants to improve service, and creating training videos to enhance citizen use [53]. We found similar results in previous literature mentioning the long-term contributions of this type of intervention in generating a critical mass of ICT (social infrastructure) users who can integrate technologies into their activities [67, 68]. Training public servants, citizens, and key stakeholders is also important for AEC [68, 71]. Montealegre [69] presented how IT can be successfully adopted even in less developed countries with contextual imperfections and scarcities. However, local contexts and traditions are essential for understanding how Colombia develops e-government [108].

Notably, we found the type of services to be a strong predictor in AEC. Thus, we found different behavior in AEC for each service; e-filings had higher levels of adoption. Also, the factors predicting each service were different. This result is consistent with Vrček and Klačmer [10], who demonstrated that 82% of the people in Croatia did not oppose less sophisticated electronic services (e.g., information-gathering), and 54% were willing to use e-government services with medium complexity (e.g., citizens sending information to the government). However, only 32% were disposed towards using e-government services for payment activities.

Overall, we found that government has a strong influence on AEC. While governments are responsible for the supply-side of e-government [11, 46], they must also play an essential role in the demand-side. Particularly, they can intervene to shape AEC's potential [11, 109]. We found that the government has important tools to manage AEC, specifically the power to establish laws and regulations that shape services and define AEC guidelines. Furthermore,

governments can configure internal aspects to conform to citizens' expectations of e-government. Additionally, governments can deploy (directly or indirectly) the knowledge required to improve e-services and increase the number of citizens using ICT. Finally, they can promote e-government services, building awareness of their existence and benefits.

Additionally, we consider that AEC should be managed mainly by government entities, as they are responsible for simplifying service processes, integrating relevant entities, offering quality e-services, creating an adequate environment for AEC (e.g., accessibility, norms and regulations, knowledge, removing barriers for adoption), promoting awareness of e-government and the multiple benefits related to its use, monitoring e-government performance and citizen satisfaction, and continuously improving services by incorporating new technologies and good practices.

## Conclusions

This research expands the knowledge of the adoption of e-government by citizens (AEC) using an institutional framework to study the government actions to increase AEC and answer our research question: how does government influence AEC? Consequently, we conducted a case study that analyzed the actions of the Antioquia Government in Colombia to increase AEC for annual vehicle tax filing and payment services and determine which of these actions could be used as AEC predictors. We employed institutional theory, institutional interventions, and legitimation strategies to classify these actions presenting four hypotheses to identify if coercive pressure, conformance to the environment, mobilization, and knowledge deployment predict AEC. Thus, we analyzed the correlation of these actions with AEC for the two services from 2015 to 2020.

We found that governments have a strong influence on AEC. First, the government controls coercive pressure, an essential factor in predicting e-filings and e-payment services. For e-payments, different elements of coercive pressure predicted AEC: laws and regulations establishing the deadlines, discounts, penalties, and access restrictions during the COVID-19 lockdown in 2020 that forced new citizens to use the e-services. For e-filings, only the laws and regulations (due dates) had predictive power. Second, governments can configure their internal components to conform to citizens' expectations for the quality of technical solutions and services regarding conformance to the environment. We identified conformance to the environment variable (representing software updates) predicting both services. Thus, improving technical solutions and monitoring satisfaction, ease of use, and effectiveness can help increase conformance with citizens' expectations of the technical solution quality and service quality. We concluded that no law, regulation, or mobilization could influence AEC increase without improving the quality of technical solutions and services.

Third, we found evidence of the importance of knowledge deployment in AEC in three government actions: delegating experts for operating services, training public servants to improve services, and creating training videos to enhance their usefulness for citizens. Although we could not test this statistically, based on the evidence gathered in the different interviews, we argue that this type of intervention would, in the long term, contribute to generating a critical mass of ICT (social infrastructure) users with the ability to integrate the technologies into their activities. Fourth, we found that mobilization was not significant for any service. We determined that this result was related to the nature of the studied services (yearly mandatory services, and most users are recurrent users). We consider promotion campaigns important in the early stages of adoption, creating awareness among citizens and influencing the intention to use. Finally, we found the type of service to be a crucial determinant of AEC; the levels of adoption and the factors predicting AEC were different.

This study makes significant theoretical contributions to the literature by proposing a new perspective to understand the development of AEC and by exploring the influence of institutional aspects on individuals. Analyzing the effects of the COVID-19 measures in AEC and comparing AEC for two different services offered on the same platform are important contributions. Developing insights in a Latin American context is another significant contribution. Finally, this study makes practical contributions identifying primary factors predicting AEC in Latin America that governments from developing countries can find helpful for policy development and prioritization.

This study has some limitations. First, we selected variables for the proposed model from the available information. Future research should explore additional institutional factors to complement our findings. Moreover, this work was limited to the Department of Antioquia and a specific population (car owners); therefore, caution should be exercised in generalizing the findings. Future research should explore different services using the institutional framework in other regions. Finally, in this work, we focused on identifying the predictor of AEC more than defining a model. Future researches could include institutional variables in existing or new models to study AEC.

## Supporting information

**S1 Appendix. Government actions related to the services.** Note. Promoting e-government for the current term (PE-CT); Promoting e-government for past due debts (PE-PDD); Promoting physical office for past due debts (PPO-PDD); Conform to the environment (C), mobilization (M), and laws and regulations (L).
(DOCX)

**S1 Data.**
(XLSX)

**S1 Scripts.**
(DOCX)

## Acknowledgments

The authors would like to thank CTG at Albany University for their support and guidance and the University of Medellín for its ongoing support and advice for this research. Finally, the authors would like to thank the editor and the anonymous reviewers for their constructive comments and suggestions for improvement on earlier versions of this paper.

## Author Contributions

**Conceptualization:** Juan Pablo Ramirez-Madrid, Manuela Escobar-Sierra, Isaias Lans-Vargas.

**Data curation:** Juan Pablo Ramirez-Madrid.

**Formal analysis:** Juan Pablo Ramirez-Madrid, Isaias Lans-Vargas.

**Funding acquisition:** Juan Pablo Ramirez-Madrid.

**Investigation:** Juan Pablo Ramirez-Madrid.

**Methodology:** Juan Pablo Ramirez-Madrid, Manuela Escobar-Sierra.

**Project administration:** Juan Pablo Ramirez-Madrid.

**Resources:** Juan Pablo Ramirez-Madrid.

**Software:** Juan Pablo Ramirez-Madrid, Isaias Lans-Vargas.

**Supervision:** Manuela Escobar-Sierra, Isaias Lans-Vargas, Juan Manuel Montes Hincapie.

**Validation:** Manuela Escobar-Sierra, Isaias Lans-Vargas, Juan Manuel Montes Hincapie.

**Visualization:** Juan Pablo Ramirez-Madrid.

**Writing – original draft:** Juan Pablo Ramirez-Madrid.

**Writing – review & editing:** Juan Pablo Ramirez-Madrid.

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
