## [Decision Letter · Decision Letter 0]

15 Nov 2021

PONE-D-21-33977Government influence on e-government adoption by citizens in Colombia: Empirical evidence in a Latin American contextPLOS ONE

Dear Dr. Juan Pablo,

Thank you for submitting your manuscript to PLOS ONE. After careful consideration, we feel that it has merit but does not fully meet PLOS ONE’s publication criteria as it currently stands. Therefore, we invite you to submit a revised version of the manuscript that addresses the points raised during the review process.

We look forward to receiving your revised manuscript.

Kind regards,

Rogis Baker, Ph.D

Academic Editor

PLOS ONE

Journal Requirements:

Reviewers' comments:

Reviewer's Responses to Questions

**Comments to the Author**

1. Is the manuscript technically sound, and do the data support the conclusions?

Reviewer #1: Yes

Reviewer #2: Partly

Reviewer #3: Partly

Reviewer #4: Yes

2. Has the statistical analysis been performed appropriately and rigorously? 

Reviewer #1: No

Reviewer #2: No

Reviewer #3: No

Reviewer #4: Yes

3. Have the authors made all data underlying the findings in their manuscript fully available?

Reviewer #1: Yes

Reviewer #2: Yes

Reviewer #3: Yes

Reviewer #4: Yes

4. Is the manuscript presented in an intelligible fashion and written in standard English?

Reviewer #1: Yes

Reviewer #2: Yes

Reviewer #3: Yes

Reviewer #4: Yes

5. Review Comments to the Author

Reviewer #1: The manuscript is well conceptualized in the manner that it highlights the importance of the E-Government adoption by citizens. However, the organization and structure of the paper does not provided an appetite for a reader to continue reading. the introduction of the paper is very long and has carried so many information that could be summarized. we suggest the author to use IMRAD approach to re-organize this work. literature part can be omitted or merged with the introduction.

The methodology of the study is not explanatory enough to know exactly what was done specifically in quality control, number of interviews, data analysis.

I have found that data analysis have been presented as a part of the result, i thought this could be seen in the methodological part.

Discussion should be done basing on the key areas of the study. i suggest that the discussion being guided by the central question/gap of the study so that we establish if the study answers what was motivated a researcher to conduct a study.

Conclusion should be aligned with the discussion and findings of the study which all are informed by the objective of the study

Reviewer #2: The paper discusses an interesting topic and explores a problematic that is increasingly relevant due to the global pandemic. The current version of the paper shows an important improvement compared to the previous one. New sections provide a more accurate theoretical context to the research. However, there are several problems with the paper listed below, in no particular order:

1. The paper is well written, but the structure of the empirical analysis is not clear.

2. Authors suggest that “Multiple filings for one payment are typical” which brings up the question of the number of e-fillings written by unique users or institutional users. At least some information and discussion on this topic is needed.

3. In addition, the second dependent variable (e-payment) seems to be associated with the main one. The authors claim that “Tax payment requires a previous tax filing, and e-payment requires e-filing”, so, technically there is a selection process that affects the estimation method or the population of analysis at least in the case of the e-payment.

4. Regarding the creation of the set of independent variables: Authors conducted several interviews with the service operation manager and staff, however, results are neither provided nor explained. The explanation and analysis of these interviews are crucial to understand the construction and accuracy of the independent variables.

5. The operationalization of the independent variables needs more explanation or at least some descriptive statistics. For instance, authors use two measures of environment conformance at the same time (see lines 379 and 385) without discussing the consequences of this decision and therefore making the role of each variable unclear. Another example is the use of the mobilization variable, which is the net between policies regardless of the strength of the policy.

6. The econometric analysis is limited due to the time series structure of the data. Simple regressions might provide biased results. Authors should provide more information regarding the selection of the econometric model and their implications regarding the characteristics of the data.

7. Discussion focused on comparisons with some developed countries, which is useful, but it needs to be complemented with other cases of developing countries with similar characteristics.

Reviewer #3: - In this current form the paper is really underdeveloped on the analysis part. The balance of the article leans heavily towards the literature review and institutional literature. The methods section should be more robust as all we learn about the actual analysis carried out is that: "multiple linear regression for data correlation analysis" (line 342 and 422)

- The data used for the regression is time series data and using pooled OLS as an estimator will yield biased results. The figures of the data clearly show seasonality and trend which needs to be addressed. The standard error correction should also reflect this fact (e.g.: Newey-West estimator, report unit root tests)

- The interpretation of the estimated coefficients are also problematic. In case of the year, as it is a numerical variable, it represents a time trend (and not that individual years are strong predictors).

- The method and analysis part should see major revisions, including a more detailed explanation of the regression approach used (and the merits of it given the data); robustness checks are also missing (different models, different estimators should point towards similar outcomes for the results to be considered robust and not just accidental), the authors should make efforts showing that all the relevant control variables are included.

- In particular, the construction of the laws and regulations variable is not transparent, the reader has little to work with as to how laws were coded as influental or not. Who measured it, how is influence defined, and how can a law have weekly influence? These questions are all the more important as this variable is statistically significant (altough with the flawed pooled OLS estimation).

- Based on the actual analysis carried out, I'm not sure that the lenghty literature review is neccesary (same stands for the Table 1). If that part is vital for the analysis then it should reflect on the enumerated institutional factors.

Reviewer #4: Thank you for the opportunity to read this paper. My comments are below:

- Scope of the article seems a bit niche, but generally the framework is tightly presented and the data collected is impressive and speaks to the questions asked.

- Table 3 is very difficult to parse; please re-format at the least to have Q3 take up equal space as previous cells.

- In Table 4 and 5, please include sample sizes; also I would rather see the significant p-values highlighted / boldened, than the non-significant ones.

- In the absence of any discussion of a causal inference strategy, I would be careful to avoid causal language. For example, on page 25, you use "determines"; in the abstract you also say "influencing". Please change these to "predicts/predicting" or similar language; or provide some rationale as to why the multiple regressions analysis can be interpreted causally.

6. PLOS authors have the option to publish the peer review history of their article (what does this mean?). If published, this will include your full peer review and any attached files.

Reviewer #1: No

Reviewer #2: No

Reviewer #3: No

Reviewer #4: No

---

## [Author Response · Author response to Decision Letter 0]

7 Jan 2022

29 December 2021

Prof. Rogis Baker, 

Academic Editor

PLOS ONE

Dear Editor:

We are submitting a revised version of the manuscript with number PONE-D-21-33977 entitled “Government influence on e-government adoption by citizens in Colombia: Empirical evidence in a Latin American context” for its acceptance for publication. We appreciate and thank the reviewers for providing their comments that have considerably improved the quality and presentation of our work.

A point-by-point answer to the referees’ comments is presented below:

Reviewer #1: 

The manuscript is well conceptualized in the manner that it highlights the importance of the E-Government adoption by citizens. However, the organization and structure of the paper does not provided an appetite for a reader to continue reading. the introduction of the paper is very long and has carried so many information that could be summarized. we suggest the author to use IMRAD approach to re-organize this work. literature part can be omitted or merged with the introduction.

Answer 1: Thank you very much for your time and valuable comments that helped us improve this new version. According to this recommendation, we have updated this new version to summarize and merge the literature review with the introduction. We have also suppressed Table 1 that lists all the institutional factors identified. Instead, we only mentioned the theoretical referents that we used to define our hypothesis in the introduction (page 3, line 69). 

The methodology of the study is not explanatory enough to know exactly what was done specifically in quality control, number of interviews, data analysis.

Answer 2: We have updated de data and methods subsection (page 11, line 260) of the methodology section detailing the process we followed for the methodology. 

For example, we indicated in the first paragraph the following (page 11, line 261):

“This is an explanatory case study [86,87] that used interviews and documentation analyses combined with quantitative analyses of the historical data from 2015 to 2020 (more than 11 million filing records and more than 5 million payments, as illustrated in Table 1) to identify how the government influences AEC (our dependent variable). We employed descriptive methods, including aggregated results for the use of e-government services in graphical and tabular output [88,89]. Additionally, we conducted multiple regressions for the data correlation analysis. Similar approaches have been used in previous research related to AEC [90,91].”

We mentioned that we had 27 interviews, and what we did in the different interviews. For example (page 13, line 301): “Finally, we used six sessions to discuss and confirm the results and prepare the report. All the data from different sources support the credibility of the findings as they allow triangulation and capture contextual complexity [94].”

I have found that data analysis have been presented as a part of the result, i thought this could be seen in the methodological part.

Answer 3: We moved the data analysis to identify the actions performed by the government to increase AEC to the methodology section. Additionally, we included the reference to Appendix A that was unintendedly omitted in the last version, despite the file being uploaded.

Discussion should be done basing on the key areas of the study. i suggest that the discussion being guided by the central question/gap of the study so that we establish if the study answers what was motivated a researcher to conduct a study.

Answer 4: This research expands the knowledge of AEC using an institutional framework to study the government actions to increase AEC and answer our research question (how does government influence AEC?). We also presented four hypotheses to identify if coercive pressure, conformance to the environment, mobilization, and knowledge deployment predict AEC.

At the end of the discussion section, we answered the research question in the following paragraph (page 25, line 568): “Overall, we found that government has a strong influence on AEC. While governments are responsible for the supply-side of e-government [11,46], they must also play an essential role in the demand-side. Particularly, they can intervene to shape AEC’s potential [11,109]. We found that the government has important tools to manage AEC, specifically the power to establish laws and regulations that shape services and define AEC guidelines. Furthermore, governments can configure internal aspects to conform to citizens’ expectations of e-government. Additionally, governments can deploy (directly or indirectly) the knowledge required to improve e-services and increase the number of citizens using ICT. Finally, they can promote e-government services, building awareness of their existence and benefits.” 

We also presented the result for the four hypotheses for both services in Table 6 at the beginning of the discussion section (page 22, line 483).

Conclusion should be aligned with the discussion and findings of the study which all are informed by the objective of the study

Answer 5: We adjusted the conclusion according to your comment. We removed the third paragraph from this section, referring to how AEC should be managed. We also expanded the initial paragraph of this section as follows (page 26, line 587): “This research expands the knowledge of the adoption of e-government by citizens (AEC) using an institutional framework to study the government actions to increase AEC and answer our research question: how does government influence AEC? Consequently, we conducted a case study that analyzed the actions of the Antioquia Government in Colombia to increase AEC for annual vehicle tax filing and payment services and determine which of these actions could be used as AEC predictors. We employed institutional theory, institutional interventions, and legitimation strategies to classify these actions presenting four hypotheses to identify if coercive pressure, conformance to the environment, mobilization, and knowledge deployment predict AEC. Thus, we analyzed the correlation of these actions with AEC for the two services from 2015 to 2020.”

Reviewer #2: 

The paper discusses an interesting topic and explores a problematic that is increasingly relevant due to the global pandemic. The current version of the paper shows an important improvement compared to the previous one. New sections provide a more accurate theoretical context to the research. However, there are several problems with the paper listed below, in no particular order:

1. The paper is well written, but the structure of the empirical analysis is not clear.

Answer 6: Thank you very much for your time and valuable comments that helped us improve this new version. As we mentioned in answer 2, we have updated the methodology section to clarify the empirical analysis.

2. Authors suggest that “Multiple filings for one payment are typical” which brings up the question of the number of e-fillings written by unique users or institutional users. At least some information and discussion on this topic is needed.

Answer 7: At the end of the case study sub-section, we included some examples to clarify this situation, expanding the discussion and giving examples. We have the following paragraph (page 11, line 247): “Tax payment requires a previous tax filing, and e-payments require e-filings. Multiple filings for one payment are typical. Also, having filings without payment is usual. For example, a tax filing to pay with discounts should be updated if not paid before the deadline. Another example is presented when the vehicle’s information should be updated (e.g., the new owner or updated value). Thus, the payment is performed based on the last tax filing. It is impossible to pay through e-government channels once the tax is filed physically in an office, and this means a selection process affects e-payments defined by the filing presented by the electronic channels.”

3. In addition, the second dependent variable (e-payment) seems to be associated with the main one. The authors claim that “Tax payment requires a previous tax filing, and e-payment requires e-filing”, so, technically there is a selection process that affects the estimation method or the population of analysis at least in the case of the e-payment.

Answer 8: In complementing answer 7, we recognized this in the same sub-section. In addition, we discussed this situation, and considering that e-filing has high levels of adoption, we concluded that the change in the size of the population for e-payment will not be significant for our analysis. In addition, we have defined how to measure AEC for each service (page 12, line 277): “We concluded that our AEC measurement would be the number of services requested in the e-government channel divided by the total number of services requested (e-payments/total payments and e-filings/total filings).”

4. Regarding the creation of the set of independent variables: Authors conducted several interviews with the service operation manager and staff, however, results are neither provided nor explained. The explanation and analysis of these interviews are crucial to understand the construction and accuracy of the independent variables.

Answer 9: In the data and methods subsection (page 11) and in the government actions to increase AEC sub-sections (page 13), we included the explanation and analysis of these interviews and how we defined the variables. We also included the reference to Appendix A that was unintendedly omitted in the last version, despite the file being uploaded.

For example, to define the variable conformance to the environment, we included the following paragraph (page 14, line 322): “Second, we found several technical solution updates to the e-services since 2017. We analyzed the objectives of these updates, and they refer to software improvements, mainly for security, privacy, usability, information quality, and new functionalities. We classified this software update as conformance to the environment [62]. To define an independent variable, we used a cumulative variable that added one to the value for every software update in the corresponding week. The objective was to reflect the solution’s maturity accumulated over time.”

5. The operationalization of the independent variables needs more explanation or at least some descriptive statistics. For instance, authors use two measures of environment conformance at the same time (see lines 379 and 385) without discussing the consequences of this decision and therefore making the role of each variable unclear. Another example is the use of the mobilization variable, which is the net between policies regardless of the strength of the policy.

Answer 10: As we mention in answer 9, we have expanded the operationalization process of the independent variables.

6. The econometric analysis is limited due to the time series structure of the data. Simple regressions might provide biased results. Authors should provide more information regarding the selection of the econometric model and their implications regarding the characteristics of the data.

Answer 11: We expanded the statistical analysis sub-section (page 18). This version presented the different models used for the analysis and explained the selection process. In addition, in this section, we explained the implications regarding the characteristics of the data. For example, we have included the following paragraph (page 18, line 419): “We identified in Table 4 that, from the five variables, only mobilization did not predict AEC for e-payment with a p=0.636. However, OLS regression has favorable properties if its assumptions are met but can give misleading results if those assumptions are not met. Thus, OLS is not robust to violations of its assumptions (normality and homoscedasticity). In this regression, we found that errors were not normally distributed across the data (Prob. Omnibus = 0.000) and heteroscedasticity in the variance of the errors across the dataset (Durbin-Watson: 0.639). This situation is typical of regressions applied to time series data, like our case.”

On page 19 (line 433) we have the following text: “…Consequently, we executed robust OLS for heteroscedasticity and autocorrelation consistency (HAC) regression. Robust regression methods are designed to be not overly affected by violations of the assumptions. Results will be presented in Tables 6 and 7.”

Finally, on page 20, we have: “Complementarily, we expanded the analysis with a different regression model to verify previous results. Thus, we conducted a generalized linear regression (GLS): a Tweedie family distribution configured as a Poisson and Gamma distribution compound. In the GLS, errors can follow any distribution of the exponential family, and homoscedasticity is not essential for the distribution of the errors. Consequently, Table 8 presents the GLS regression results for AEC e-payment…”

7. Discussion focused on comparisons with some developed countries, which is useful, but it needs to be complemented with other cases of developing countries with similar characteristics.

Answer 12: We have included discussions on Latin American and Asian developing countries, and including references to Colombia.

For example, on page 23 (line 511), we included the following paragraph: “Indeed, Latin American governments have an essential role in combating the factors that widen or retain the digital divide, especially in rural areas and developing countries [102]. Increasing access to computers and the Internet (as the Colombian government promotes) is not a complete solution but a good start nonetheless [103]. The Internet can promote citizenship and citizen participation in Latin America [104]. Also, improving websites’ accessibility to avoid discrimination, even imposing sanctions for non-compliance of standards as another example of coercive pressure, could be another strategy followed by developing countries [78].”

Another example is presented on page 24: “… Training public servants, citizens, and key stakeholders is also important for AEC [68,71]. Montealegre [69] presented how IT can be successfully adopted even in less developed countries with contextual imperfections and scarcities. However, local contexts and traditions are essential for understanding how Colombia develops e-government [108].”

 

Reviewer #3: 

- In this current form the paper is really underdeveloped on the analysis part. The balance of the article leans heavily towards the literature review and institutional literature. The methods section should be more robust as all we learn about the actual analysis carried out is that: "multiple linear regression for data correlation analysis" (line 342 and 422).

Answer 13: Thank you very much for your time and valuable comments that helped us improve this new version. We have updated the article’s structure (please see answer 1) for a better balance. Also, we have updated our methodology section to make it more robust (please see answers 2 and 9).

- The data used for the regression is time series data and using pooled OLS as an estimator will yield biased results. The figures of the data clearly show seasonality and trend which needs to be addressed. The standard error correction should also reflect this fact (e.g.: Newey-West estimator, report unit root tests)

Answer 14: You are right. Consequently, we have updated the statistical analysis sub-section, as we mentioned in answer 11.

- The interpretation of the estimated coefficients are also problematic. In case of the year, as it is a numerical variable, it represents a time trend (and not that individual years are strong predictors).

Answer 15: You are right. Initially, we have identified this variable as an indicator of the system's maturity through time. However, according to your comment, we reviewed the model and decided to remove the year as a predictor of AEC. For the maturity of the system, we kept the conformance to the environment variable.

- The method and analysis part should see major revisions, including a more detailed explanation of the regression approach used (and the merits of it given the data); robustness checks are also missing (different models, different estimators should point towards similar outcomes for the results to be considered robust and not just accidental), the authors should make efforts showing that all the relevant control variables are included.

Answer 16: As we mentioned in Answers 11 and 14, we expanded our explanation about the statistical analysis indicating what analyses we conducted and why we decided to use that type of regression. Additionally, we mentioned that we executed three separate regressions with consistent results. Also, we identified the type of service as a relevant control variable in the available data, and we decided to run separated models for each service.

- In particular, the construction of the laws and regulations variable is not transparent, the reader has little to work with as to how laws were coded as influental or not. Who measured it, how is influence defined, and how can a law have weekly influence? These questions are all the more important as this variable is statistically significant (altough with the flawed pooled OLS estimation).

Answer 17: Expanding on Answer 9, we included details on how we defined this variable. In the sub-section actions performed to increase AEC (page 13, line 310), we included the following paragraph: “First, the government establishes two payment due dates every year. The first date allows payment due at a 10% discount, and the second date is for the amount due without penalties, but interest accrues after this date. We also found some laws and regulations establishing special discounts in specific periods. Initial analyses showed that citizens that pay with discounts have higher AEC levels than citizens that pay with penalties. Also, we found that weeks with due dates have presented an increase in AEC, possibly since physical offices cannot receive the demand in these periods. Consequently, we classified these actions as coercive pressure [51], defining two variables: Payment and laws and regulations. We used the variable payment to establish the value of paying; 0.8 for the 20% discount period, 0.9 for the 10% discount period, 1.0 for the no-discount period, and 1.0 plus monthly interest rate for the weeks after the due date. For the variable laws and regulations, we used 1 if there was a deadline date in that week and 0 if it did not.”

- Based on the actual analysis carried out, I'm not sure that the lenghty literature review is neccesary (same stands for the Table 1). If that part is vital for the analysis then it should reflect on the enumerated institutional factors.

Answer 18: You are right. Please see Answer 13.

 

Reviewer #4: 

Thank you for the opportunity to read this paper. My comments are below:

- Scope of the article seems a bit niche, but generally the framework is tightly presented and the data collected is impressive and speaks to the questions asked.

- Table 3 is very difficult to parse; please re-format at the least to have Q3 take up equal space as previous cells.

Answer 19: Thank you very much for your time and valuable comments that helped us improve this new version. We have decided to create a new table (Table 3) for Q3, as you can see on page 15 (line 348). Q1 and Q2 remain in Table 2 (line 345).

- In Table 4 and 5, please include sample sizes; also I would rather see the significant p-values highlighted / boldened, than the non-significant ones.

Answer 20: We have included the sample sizes, and we have boldened the significant p-values instead of the non-significant ones. 

- In the absence of any discussion of a causal inference strategy, I would be careful to avoid causal language. For example, on page 25, you use "determines"; in the abstract you also say "influencing". Please change these to "predicts/predicting" or similar language; or provide some rationale as to why the multiple regressions analysis can be interpreted causally.

Answer 21: We have updated our manuscript, avoiding causal language, and consequently defined our hypothesis. For example, in the introduction, we have the following paragraph (page 6): “The legal framework, as an example of coercive pressure, is determinant in the context of AEC [54]. Several studies have confirmed that the government’s entities use this coercive pressure to encourage citizens and other groups to embrace e-government [13,49,55–58], even in mandatory use contexts, as Denmark and the UK have defined [59]. Consequently, we have defined that coercive pressure predicts AEC as our first hypothesis.”

Additionally, we included a table in the discussion section consolidating the hypothesis results (page 22, line 483). There you can confirm that we implemented your recommendation.

---

## [Editor Report · Decision Letter 1]

14 Feb 2022

Government influence on e-government adoption by citizens in Colombia: Empirical evidence in a Latin American context

PONE-D-21-33977R1

Dear Dr. Juan Pablo RAMIREZ-MADRID,

We’re pleased to inform you that your manuscript has been judged scientifically suitable for publication and will be formally accepted for publication once it meets all outstanding technical requirements.

Kind regards,

Rogis Baker, Ph.D

Academic Editor

PLOS ONE
---

## [Editor Report · Acceptance letter]

16 Feb 2022

PONE-D-21-33977R1 

Government influence on e-government adoption by citizens in Colombia: Empirical evidence in a Latin American context 

Dear Dr. Ramirez-Madrid:

I'm pleased to inform you that your manuscript has been deemed suitable for publication in PLOS ONE. Congratulations! Your manuscript is now with our production department. 

Kind regards, 

on behalf of

Dr. Rogis Baker 

Academic Editor

PLOS ONE